# Entomopathogenic Fungi as a Potential Management Tool for the Control of Urban Malaria Vector, *Anopheles stephensi* (Diptera: Culicidae)

**DOI:** 10.3390/jof9020223

**Published:** 2023-02-08

**Authors:** Siddaramegowda Renuka, Chalageri Vani H, Eapen Alex

**Affiliations:** 1ICMR-National Institute of Malaria Research Field Unit, Bengaluru 562110, Karnataka, India; 2ICMR-National Institute of Malaria Research Field Unit, Chennai 600077, Tamil Nadu, India

**Keywords:** *Anopheles stephensi*, *Beauveria bassiana*, *Metarhizium anisopliae*, WHO cone bioassay, Entomopathogenic fungi, vector control

## Abstract

*Anopheles stephensi* (Diptera: Culicidae) is the vector of urban malaria in India and has a significant impact in transmitting infection in cities and towns. Further, WHO has also alarmed its invasive nature as a threat to African countries. Entomopathogenic fungi such as *Beauveria bassiana* and *Metarhizium anisopliae* have been found to be highly effective in controlling vector mosquito populations and therefore could be used in integrated vector control programs. Before employing the entomopathogenic fungi into the control programs, an effective isolate must be selected. Two separate experiments were conducted to evaluate the efficacy of *Beauveria bassiana* (Bb5a and Bb-NBAIR) and *Metarhizium anisopliae* (Ma4 and Ma-NBAIR) isolates against *An. stephensi*. Cement and mud panels were treated with fungal conidia with the concentration of 1 × 10^7^ conidia/mL and adult *An. stephensi* mosquitoes were exposed to the treated panels (24 h after conidia were applied) by conducting WHO cone bioassay tests. The survival of the mosquitoes was monitored daily until the 10th day. In the second experiment, second instar larvae of *An. stephensi* were treated with fungal (Bb5a, Bb-NBAIR, Ma4 and Ma-NBAIR) conidia and blastospores with the spore concentration of 1 × 10^7^ spores/mL. The survival of larvae was monitored until pupation. All the fungal isolates tested caused mortality in the adult mosquitoes, with varying median survival times. The Bb5a isolate reported lesser median survival times on both cement and mud panels (6 days). The treated mosquitoes showed similar survival rates for each fungal isolate irrespective of the panel type. There was no mortality in the treated larvae; however, a delay in larval development to pupae was observed compared with the untreated control larvae. Ma4-treated larvae took 11 days (95% CI = 10.7–11.2) to become pupae when compared with the untreated control larvae (6 days [95% CI = 5.6–6.3]). The findings of this study will be useful to consider EPF as a tool for the management of vector mosquitoes.

## 1. Introduction

Malaria—a life threatening mosquito-borne disease—has a significant impact on human health worldwide [1]. The World Health Organization (WHO) reported 241 million malaria cases and 627,000 malaria deaths worldwide in 2020 [2]. The control of malaria can be achieved either by controlling the *Plasmodium* (the parasite) or through vector mosquitoes. However, *Plasmodium* exhibits a lesser effectiveness towards the wide range of currently available malaria drugs because of its resistance [3,4,5]. On the other hand, most of the primary malaria vector species of the genus *Anopheles* have shown resistance towards insecticides. The use of insecticide-treated bed nets (ITNs) and indoor residual spraying (IRS) are highly effective against vector mosquitoes, but their efficacy is threatened by the emergence of resistance towards synthetic insecticides [6]. The evolutionary adaptation of the mosquitoes to the insecticides may be due to the frequent application of insecticides such as organochlorines, organophosphates (OPs), carbamates and pyrethroids. The increased selection pressure in the mosquitoes will reduce the lifespan of currently available insecticides and will increase the cross-resistance to the newly developed insecticides. Mosquitoes are also able to withstand a wide range of physical conditions due to their genetic plasticity and reproductive ability. The degree of resistance varies among mosquito species, insecticide classes and regions [7], so there is a need for alternative eco-friendly vector control measures.

The use of entomopathogenic fungi (EPF) as an alternative tool for the control of malaria vectors seems to be promising. EPF species belonging to the genera *Beauveria* and *Metarhizium* are well-studied biocontrol agents that control a wide range of insects [8,9,10,11,12,13,14]. Unlike other biocontrol agents such as bacteria, microsporidia and viruses, hypocrealean EPF infect insects through adhesion [15]. Fungal conidia attach to the insect cuticle and form the germ tube and appressorium. The appressorium produces the peg, which penetrates the insect cuticle with the help of mechanical pressure and cuticle degrading enzymes [16]. The fungal hyphae reach the haemocoel and proliferate using insect nutrients. Once the fungi overcome the host immune defenses, they kill the insect [17]. Since the EPF takes a long time to infect and kill the insects, selection pressure in the insects is unlikely. *B. bassiana* and *M. anisopliae* were employed in the control of vector mosquitoes under both laboratory and field conditions. Fungal spores can be easily applied to bed nets and curtains [18] and cement and mud surfaces [19] in the form of simple formulations using oil or water-based solutions and they can persist for a couple of months on treated surfaces [20]. Spores can also be incorporated in attracting odor traps [21]. EPF might be used as a synergy with various insecticides or alone in integrated vector management approaches. Fungal infection reduces the lifespan of both susceptible and insecticide-resistant mosquitoes [22,23]. EPF may be useful in reducing the probability of a female mosquito in host-seeking behavior and blood feeding tendency and it might also reduce the probability of a gravid female searching for a suitable oviposition site [22,24]. Blanford et al. have demonstrated the direct effect of EPF on the development of *Plasmodium* in mosquitoes using mouse malaria as a model system [23]. It was observed that only 8% of mosquitoes that were infected with both the parasite and the fungi had transmissible parasites after 14 days of fungal exposure compared with 35% of mosquitoes infected alone with the *Plasmodium*. As far as malaria is concerned, even a slight reduction in the number of bites per individual mosquito will reduce the risk of malaria transmission [25].

Currently, the adult vector control interventions in operation are mainly IRS and LLIN, both having chemical compounds (pyrethroids that are partially accepted by the community because of allergic reactions to the skin when in contact and upper respiratory discomfort/illness as well, especially in elderly people with co-morbidities). EPF is environmentally safe, eco-friendly and economically cheaper than pyrethroid compounds, with less hazards to people. In addition, in places where IRS is not accepted, suitable effective formulations of EPF can be used against malaria vectors, even in urban areas, unlike IRS. However, the aim of the present study was to evaluate the pathogenicity of EPF against immature and adult female *An. stephensi* mosquitoes. Exposure of EPF to infect and kill the mosquitoes is an important consideration while selecting the site for fungal application. Cement and mud panels were selected for conidial application in the current study to represent a realistic exposure site, since the adult mosquitoes spend 15–30 min on wall surfaces inside the house when they seek a blood meal and up to 24 h after blood feeding. Studies were carried out in two separate experiments.

## 2. Materials and Methods

### 2.1. Institutional Ethical Clearance (IEC)

The present work does not involve any human or animal trials. However, Institutional Animal Ethics Committee (IAEC) clearance of the project was obtained from the Indian Council of Medical Research-National Institute of Malaria Research, New Delhi with the CPCSEA registration No.33/Go/ReBi/S/99/CPCSEA, dated on 20 April 2022.

### 2.2. Maintenance of An. stephensi Colonies

Immature mosquitoes of *An. stephensi* were collected from Devanahalli, Bangalore, India and were maintained in the insectary ICMR-National Institute of Malaria Research Field Unit (ICMR-NIMRFU), Bangalore according to WHO standard protocols [26]. Mosquitoes were identified based on standard identification keys [27]. The larvae were maintained in plastic trays and were fed on Tetramin**^®^** baby fish food until pupation. The pupae were collected in a plastic cup with water and were transferred into the cage. The adults were maintained in the cages with 10% glucose (*w*/*v*) solution. For female mosquitoes that were blood feeding, live Swiss albino mice were used as a blood source, approved by the Institutional Animal Ethics Committee (IAEC)-ICMR-NIMR, New Delhi, with the CPCSEA registration No.33/Go/ReBi/S/99/CPCSEA. Three-to-five-day-old adult female mosquitoes emerged, were fed with 10 percent glucose and were used for the bioassays. During the bioassays, mosquitoes were maintained with 10% glucose solution.

### 2.3. Fungal Isolates

*B. bassiana* isolates, namely Bb5a and Bb-NBAIR (referenced), and *M. anisopliae* isolates, namely Ma4 and Ma-NBAIR (referenced), were procured from ICAR-NBAIR, Bangalore (Table 1). These fungal isolates were originally isolated from soil and insect cadaver samples. The fungal isolates were sub-cultured on Sabouraud’s Dextrose Yeast Agar (SDYA) (dextrose 40 g, mycological peptone 10 g, yeast extract 5 g, agar-agar 15 g in 1000 mL of distilled water) medium and were stored at 4 °C for further studies.

### 2.4. Experiment 1

The bioassay was undertaken to evaluate the pathogenicity of Bb5a, Bb-NBAIR, Ma4 and Ma-NBAIR fungal isolates against female adult *An. stephensi* mosquitoes.

#### 2.4.1. Preparation of Conidial Suspension

*B. bassiana* (Bb5a and Bb-NBAIR) and *M. anisopliae* (Ma4 and Ma-NBAIR) isolates were inoculated into respective agar plates containing SDYA. Agar plates were incubated at 25 ± 1 °C for 15 days. After incubation, conidia were harvested by scraping the sporulated colonies on the surface of the agar and suspended into respective conical flasks containing sterile filter water with 0.05% Tween 80 (*v*/*v* aqueous solution). The resulting conidial suspensions were filtered separately through sterile filter paper (Whatman No.1) to obtain hyphal free conidial suspension. The conidial concentration in the conidial suspension was determined using a haemocytometer and the desired concentration of 1 × 10^7^ conidial/mL was adjusted using sterile filtered water [28]. Conidial viability in the conidial suspension was measured at the beginning of each experiment. Approximately 100 µL of conidial suspension was pipetted out from the above-prepared conidial suspension with a conidial concentration of 1 × 10^7^ conidial/mL and spread over SDYA agar plates and the plates were kept for incubation at 25 ± 1 °C. After 24 h of incubation, the agar plate lids were removed and a cover slip was placed over the surface of the agar and the conidia were examined under a binocular compound microscope to determine the percentage of conidial germination. A minimum of 200 conidia were examined and the percent of germination was calculated. Conidial suspension showing the viability of >90% was used for cone bioassay [29].

#### 2.4.2. Test Panel Preparation

Cement and mud panels (surface) were selected for fungal treatment in the present study since these panels were similar to the walls of the houses in most of the villages and the study was intended to target resting mosquitoes. Panels were prepared at the ICMR-NIMRFU, Bangalore. The cement panels were prepared using cement and sand in the ratio of 1:5 and the mud panels were made with red soil collected at the ICMR-NIMRFU campus, Bangalore. Cement and mud panels were 1 cm in thickness and measured about 30 × 30 cm^2^. Panels were left to dry for about 4 weeks at room temperature [30]. Cement and mud panels were treated with respective fungal conidial suspensions using locally available hand-held air compressor conventional spray guns with 40 to 60 psi and having 1.4 to 2.5 mm nozzle sizes. The sprayer was held approximately 30 cm away and at a right angle to the panel surface. About 20 mL of respective fungal conidial suspension was applied onto each of the respective cement and mud panels. The treated panels were left to dry for 24 h at ambient temperature and humidity (~20–28 °C, 40–80% RH) and were used for cone bioassay after 24 h of treatment [31].

#### 2.4.3. Cone Bioassay Test

Cone bioassay tests were performed on the treated cement and mud panels according to WHO protocol [32]. Four replicates were maintained for each treatment and control on both the panels (cement and mud). Three-to-five-day-old non-blood fed 20 female adult mosquitoes were used for each replicate. Adult female mosquitoes were exposed for 30 min on the treated panels (Figure 1). Then, the treated mosquitoes were transferred using an electronic aspirator to respective holding cups and the mosquitoes were fed with a 10% glucose solution. The mortality of the treated mosquitoes was recorded up to 10 days at 24 h intervals. Dead mosquitoes were collected, surface sterilized with 70% ethanol and kept in a sterilized petri plate lined with moist filter paper to facilitate the fungal growth [33].

### 2.5. Experiment 2

The larval susceptibility test was aimed to compare the virulence of conidia and blastospores of four fungal isolates, namely Bb5a, Bb-NBAIR, Ma4 and Ma-NBAIR, against the second instar larvae of *An. stephensi*.

#### 2.5.1. Preparation of Conidia and Blastospores Suspension

Conidial suspensions of Bb5a, Bb-NBAIR, Ma4 and Ma-NBAIR isolates were prepared as described in the above Experiment 1. Blastospores were produced in Sabouraud’s Dextrose Yeast Broth (SDYB) medium (dextrose 40 g, mycological peptone 10 g, yeast extract 5 g in 1000 mL of distilled water). A total of 25 mL of SDYB was taken in 50 mL conical flasks and sterilized at 121 °C for 15 min in an autoclave. After sterilization, the medium was inoculated with a loopful of respective 15-day-old fungal culture (Bb5a, Bb-NBAIR, Ma4 and Ma-NBAIR) and then incubated at 27 °C in a rotary shaker at 130 rpm for 7 days. After incubation, the culture medium was filtered through sterile filter paper (Whatman No.1) to remove the fungal mat. The fungal filtrate was then centrifuged at 5000 rpm for 10 min and the pellet was suspended in sterile filter water. Blastospore concentration in the suspension was determined using an improved Neubauer haemocytometer, and desired blastospore concentration 1 × 10^7^ blastospore/mL was adjusted using sterile filtered water [34].

#### 2.5.2. Larval Susceptibility Test

The assays were conducted according to WHO protocol [35] to compare the virulence of fungal conidia and blastospores against *An. stephensi*. Four replicates were maintained for each treatment and the control. A total of 25 early second instar larvae of *An. stephensi* were maintained per replicate. An amount of 1 mL of conidia and blastospore suspension of Bb5a, Bb-NBAIR, Ma4 and Ma-NBAIR was added into respective plastic cups containing 99 mL of RO filtered water. Then, 25 numbers of early second instar larvae of *An. stephensi* were released into respective plastic cups. The control cups were treated with 1 mL of sterilized water containing 0.05% aqueous Tween 80 (*v*/*v*). The larvae were fed on Tetramin**^®^** baby fish food till pupation. The assays were performed at 25 ± 2 °C.

### 2.6. Statistical Analysis

The control mortality was adjusted using Abbott’s formula [36] whernever applicable. The percent mortality and mycosis data were analyzed using analysis of variance. The least significant difference (LSD) test was used to compare the means. The Kaplan–Meier survival analysis method was used to obtain median survival times (MST) for treated and untreated groups of mosquitoes. Significant differences between the fungal species were estimated using the Log Rank (Mantel–Cox) test using SPSS version 28.0.1.1 (15).

## 3. Results

### 3.1. Experiment 1

The adult mortality (%) and mycosis (%) were observed after 10 days of treatment (Figure 2). Significant differences in the percentage mortality and in mycosis were observed among the different isolates screened (Table 2 and Table 3). A lower survival rate was observed in the fungus-treated mosquitoes than in the untreated control. The fungus mediated mortality of *An. stephensi* on treated cement panels ranged from 33.75 to 86.25% and on mud panels it was from 48.75 to 88.75%. The Bb5a-treated mosquitoes had lesser median survival times (6 days), followed by the Bb-NBAIR-treated mosquitoes (7 days) on both cement (χ^2^ = 174.04; df = 4; *p* value < 0.01) and mud (χ^2^ = 169; df = 4; *p* value < 0.01) panels, respectively (Table 4). The treated mosquitoes showed similar survival rates for each fungal isolate, irrespective of the panel type (Figure 3).

### 3.2. Experiment 2

The second instar larvae of *An. stephensi* treated with fungal conidia and blastospores indicated prolonged larval duration and took the longest median day for developing into the pupal stage in all the treatments (Table 5). Among the fungal isolates tested, the conidia of the Ma4-treated larvae took 11 days (95% CI = 10.7–11.2), followed by Bb-NBAIR (10 days [95% CI = 9.6–10.3]), Bb5a (8 days [95% CI = 7.4–8.5, and]) and Ma-NBAIR (8 days [95% CI = 7.5–8.4]) to become pupae when compared with the untreated control larvae (6 days [95% CI = 5.6–6.3]). The blastospores of the Bb5a- and Ma4-treated larvae also took 11 days (95% CI = 10.1–11.8) to become pupae. Further, the Ma-NBAIR-treated larvae took 10 days (95% CI = 9.4–10.5) and the Bb-NBAIR-treated larvae took 9 days (95% CI = 8.6–9.3) when compared with the untreated control larvae (6 days [95% CI = 5.7–6.2]) (Figure 4). The mean time required for the second instar larvae of *An. stephensi* to undergo stage change to pupa is represented in Table 5. The significant differences between the fungal species were estimated with χ^2^ = 440.23; df = 4 and *p* value < 0.01 for conidia-treated larvae and χ^2^ = 637.52; df = 4 and *p* value < 0.01 for blastospore-treated larvae using the Log Rank (Mantel–Cox) test.

## 4. Discussion

The present study was intended to screen EPF isolates against adult and larvae of the *An. stephensi* mosquito to select the promising EPF isolates for the development of formulation of EPF against malaria vectors. The tested fungal isolates exhibited reduced survival rates of the adult mosquitoes of *An. stephensi,* with varying percent mortality by the 10th day when compared with the untreated control mosquitoes. Isolate Bb5a showed the lowest survival time for *An. stephensi,* 6 ± 0.47 days (cement) and 6 ± 0.40 days (mud). Snetselaar et al. reported that the *B. bassiana* isolate was able to significantly increase mortality of the free-flying *Aedes aegypti* adults compared with the control when the gauze of the mosquito traps was contaminated with the *B. bassiana* conidia [37]. In this study, the efficacy of fungal isolates does not vary among the treated panels tested. However, a few studies have reported that variations in the efficacy of the treatment between different surfaces are not unique. The pyrethroid impregnated bed nets revealed that the efficacy is dependent on the type of fabric used; the pyrethroid-treated polyester net showed more effectiveness than the nylon and cotton ones [38]. Howard et al. reported that the pyrethroid resistance *An. gambiae* with increased susceptibility to *M. anisopliae* and *B. bassiana* when mosquitoes were exposed to impregnated polyester netting [39]. In contrast, Mnyone et al. reported the efficacy of the fungal conidia to be higher on the mud panels and the cotton cloth than on those mosquitoes exposed to conidia on polyester netting [19]. The variations in the efficacy of the fungal conidia/insecticides might be due to the texture of the treated surfaces. Since the synthetic polyester net has a smooth surface and a less-treatable surface area due to its netting pattern, it supports less conidial attachment to the surface and minimizes the conidial load on the surface, thereby reducing the mosquito’s exposure to the conidia. Natural surfaces (cotton nets and mud panels), due to their high absorption capability, absorb and withhold high quantities of conidia/insecticides on their surfaces and this may be the reason for indicating more efficacy. Clay pots were reported as attractive resting sites for mosquitoes and suitable for the spray application of formulated conidia. *M. anisopliae* conidia formulated in mineral oil sprayed inside clay pots remained infective and virulent, revealing a significant reduction in the longevity of *An. gambiae*s.s. and *An. funestus* mosquitoes [40].

The genus, *Beauveria* has a broad host range and the natural infection on mosquito larvae has been reported on *Culex tarsalis*, *Culex pipiens* and *Anopheles albimanus* [41]. Since the conidia of *Beauveria* are hydrophobic in nature, they float on the water surface and contact mosquito larvae that feed below the water surface. It has been reported that the siphon and head are important sites for infection [42]. Fungal conidia were found to be attached to larval surfaces after ingestion conidia were found within the gut but not in the hemocoel. Infected larvae showed swollen stages of the conidia in several body parts 24 h after exposure. The emergence of germ tubes was observed abundantly in the gut after 48 h of infection. The major site of infection of conidia in the larvae was found to be the perispiracular lobes. Extensive accumulations of developed blastospores were seen in the fore-, mid- and hind-gut [42,43]. Larval mortality might be due to a mechanical blockage of the tracheal trunks, leading to suffocation in the larvae or the production of catabolic enzymes from the fungi for larval tissue destruction. Since *B. bassiana* produces several catabolic enzymes for tissue destruction in terrestrial insects, it is not clear that *B. bassiana* uses the same mechanism in the aquatic insects [44,45]. The conidia and blastospores of *M. brunneum* were pathogenic to larvae of *Ae. aegypti*, *Cx. quinquefasciatus* and *An. stephensi* [34]. *Ae. aegypti* larvae treated with the blastospores and conidia of *B. bassiana* indicated 85 and 50 percent mortality, respectively, after 96 h post-exposure [46]. In the present study, it was observed that fungal conidia and blastospores did not show any pathogenicity towards the larvae of *An. stephensi*. Nevertheless, there was a delay in larval development into pupae when larvae were treated with fungal conidia and blastospores. Alkhaibari et al. also reported the significant reduction in the percentage of pupation of *An. gambiae* by 39–50% when the larvae were treated with *M. anisopliae* and *B. bassiana* under field conditions [47]. The development of the *Cx. pipiens* larvae was not affected when the larvae were treated with *B. bassiana*; however, it had an impact on pupal duration [48]. In general, *Ae. aegypti* takes 7 to 9 days to complete the larval stage, but it took 36 days when the larvae were treated with *B. bassiana*; this is epidemiologically significant in the disease transmission potential of the vector [49].

Although there are few reports of EPF virulence on *An. stephensi*, much of the fungi research has been conducted for the control of *An. gambiae* [40] and *Ae. aegypti* [22,50]. In India, the work conducted on EPF against adult *An. stephensi* mosquitoes is meager and this might be the first report on the *B. bassiana* isolate that has virulence to *An. stephensi* adults. Hitherto, the control methods for *An. stephensi* mosquitoes mainly rely on chemical insecticides. Hence, the development of other eco-friendly control methods is important due to mosquito resistance towards chemical insecticides as well as their deleterious impact on the environment.

## 5. Conclusions

In this study, EPF were found to be effective in reducing the survival rate (6 days at *p* value < 0.01) of the treated adult mosquitoes; it was also observed that the type of panel (cement and mud) does not affect the efficiency of the EPF. The development of larvae into pupae was delayed when the larvae were treated with EPF conidia and blastospores (10 and 11 days, respectively, at *p* value < 0.01). Among the EPF tested, Bb5a could be used for the effective control of adult mosquitoes and Ma4 for larval source management. EPF could be used alone for better management of mosquito immature controlling emergence to adults. Since the EPF reduces the survival rate of the adult mosquitoes, it can be integrated into the malaria control program for efficiently reducing the adult density and, thereby, the disease transmission.

## Figures and Tables

**Figure 1 jof-09-00223-f001:**
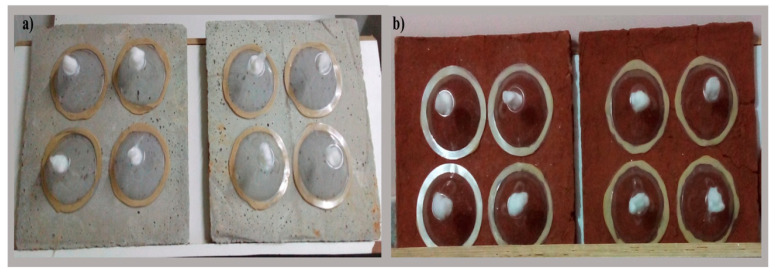
WHO cone bioassay tests were performed on EPF treated panels. (**a**) Cement panel, (**b**) mud panel.

**Figure 2 jof-09-00223-f002:**
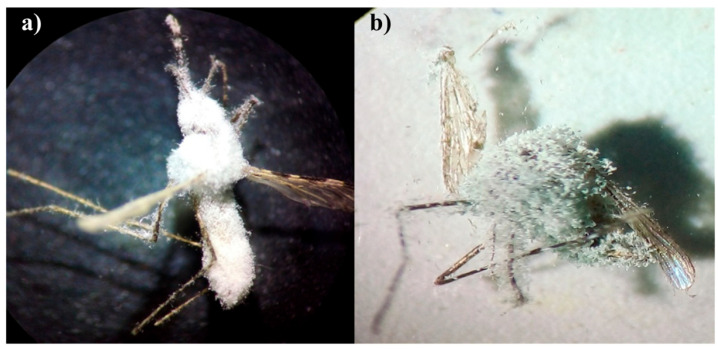
*An. stephensi* adult mosquitoes exposed to the EPF-treated panels indicating mycosis. (**a**) *An. stephensi* exposed to *B. Bassiana,* (**b**) *An. stephensi* exposed to *M. anisopliae*.

**Figure 3 jof-09-00223-f003:**
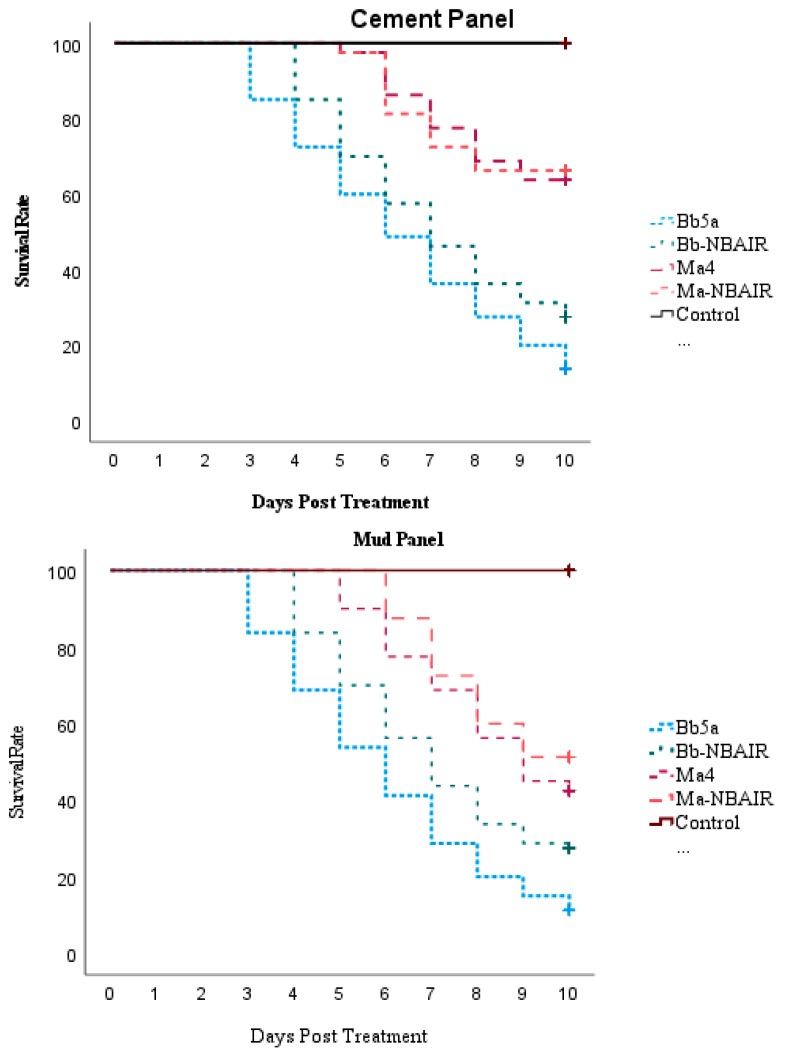
Survival rate of *An. stephensi* exposed to *B. bassiana-* and *M. anisopliae*-treated cement and mud panels through WHO cone bioassay tests.

**Figure 4 jof-09-00223-f004:**
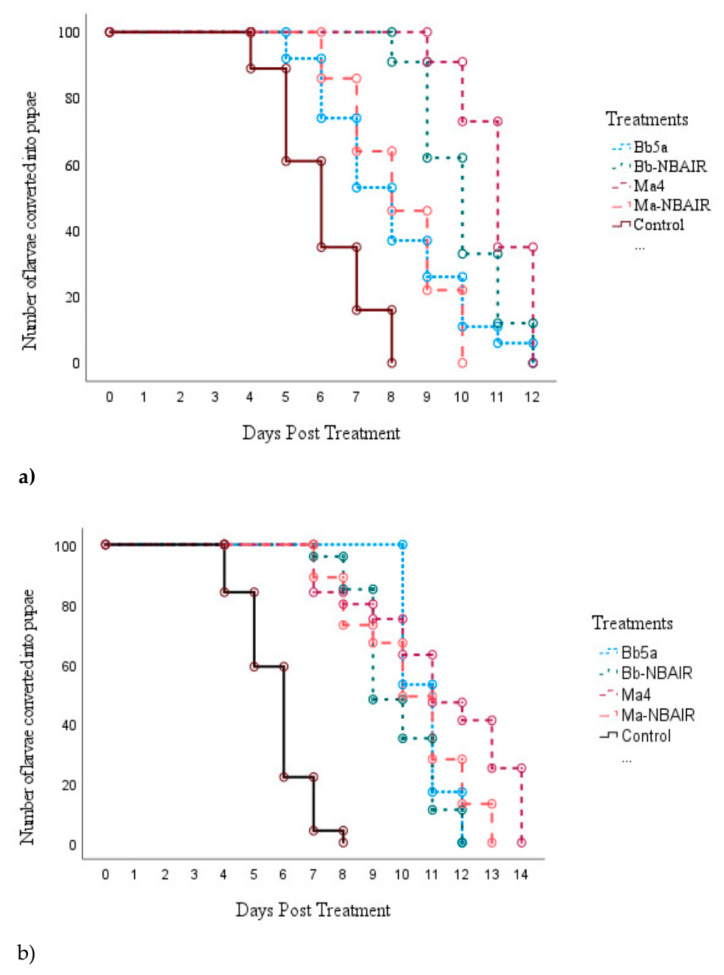
Time required for larvae of *An. stephensi* to undergo stage change to pupae when exposed to fungal conidia and blastospores. Fungal conidia (**a**) and blastospores (**b**).

**Table 1 jof-09-00223-t001:** Source details and accession number of *Beauveria bassiana* and *Metarhizium anisopliae* isolates.

EPF Isolate	Institute Isolate Code	Source of Isolation	ICAR-NBAIM Accession Number	NCBI Genbank Accession Number
Bb-5a	PDBC Bb-5a	Coffee berry borer (*Hypothenemus hampei*) cadaver	NAIMCC-F-00396	JF837134
Bb- NBAIR	PDBC Bb-3	Mottled water hyacinth weevil (*Neochetina bruchi*) cadaver	NAIMCC-F-00393	JF837139
Ma-4	PDBC Ma-4	Cashew stem and root borer (*Plocaederus ferrugineus*) cadaver	NAIMCC-F-01296	JF837157
Ma –NBAIR	PDBC Ma-15	Soil sample	NAIMCC-F-01306	JF837154

**Table 2 jof-09-00223-t002:** Impact of different isolates of *B. bassiana* and *M. anisopliae* against *An. stephensi* adult mosquitoes through cone bioassay tests on mud panels.

Mud Panel							
Mosquito Species	EPF Isolate	Percent Mortality	Mean	Standard Deviation	Percent Mycosis	Mean	Standard Deviation
*An. stephensi*	Bb5a	88.75 ^a^	44.37	4.74	86.25 ^a^	43.12	4.62
	Bb-NBAIR	72.50 ^b^	36.25	3.89	62.50 ^b^	31.25	3.34
	Ma4	57.50 ^b^	28.75	3.12	12.50 ^c^	5.62	7.76
	Ma-NBAIR	48.75 ^c^	24.37	2.63	11.25 ^c^	6.25	7.90
	Control	0.00 ^d^	-	-	0.00 ^d^	-	-
CD (*p* ≤ 0.01)		13.11			10.66		

Values represented in small alphabets (superscripted) indicate that they are significantly different from each other.

**Table 3 jof-09-00223-t003:** Impact of different isolates of *B. bassiana* and *M. anisopliae* against *An. stephensi* adult mosquitoes through cone bioassay tests on cement panels.

Cement Panel							
Mosquito Species	EPF Isolate	Percent Mortality	Mean	Standard Deviation	Percent Mycosis	Mean	Standard Deviation
*An. stephensi*	Bb5a	86.25 ^a^	43.1	4.61	83.75 ^a^	41.87	4.47
	Bb-NBAIR	72.50 ^b^	36.25	3.88	70.00 ^b^	35.00	3.76
	Ma4	36.25 ^c^	18.12	1.99	11.25 ^c^	5.62	6.78
	Ma-NBAIR	33.75 ^c^	16.87	1.86	6.25 ^c^	3.12	3.72
	Control	0.00 ^d^	-	-	0.00 ^d^	-	-
CD (*p* ≤ 0.01)		12.58			8.42		

Values represented in small alphabets (superscripted) indicate that they are significantly different from each other.

**Table 4 jof-09-00223-t004:** Median survival times (10 days post-treatment) of *An. stephensi* exposed to *B. bassiana-* and *M. anisopliae*-treated cement and mud panels through WHO cone bioassay tests.

	MST (Days)	
Fungal Isolates	Cement Panel	Mud Panel
Bb5a	6	6
Bb-NBAIR	7	7
Ma4	10	9
Ma-NBAIR	10	10
Control	10	10

**Table 5 jof-09-00223-t005:** Mean time required for 2nd instar larvae of *An. stephensi* to become pupae when exposed to conidia and blastospores of *B. bassiana* and *M. anisopliae*.

Fungal Isolates	Mean ± SE (Days)	
	Conidia	Blastospores
Bb5a	7.99 ± 0.19	10.70 ± 0.07
Bb-NBAIR	9.98 ± 0.11	9.75 ± 0.13
Ma4	10.99 ± 0.09	11.15 ± 0.25
Ma-NBAIR	8.18 ± 0.13	10.19 ± 0.19
Control	6.01 ± 0.12	5.69 ± 0.10

## Data Availability

The data presented in this study are available on request from the corresponding authors.

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
