# Peer review of "Entomopathogenic Fungi as a Potential Management Tool for the Control of Urban Malaria Vector, Anopheles stephensi (Diptera: Culicidae)"

_jof, 2023, doi:10.3390/jof9020223_

Round 1

Reviewer 1 Report

Dear Authors, 

Thanks for that study. Vector control is of high importance ( also for other diseases) and the weakening of the vector in any form should be forced and  can be a part in control programs. Screening or proofing of the most appropriate EPF (virulence, establishment, persistence, non- target effects) as a standard procedure before further research is/should be usual.

The study is described in detail, and the material and methods as well as results are clearly presented and of importance for the specific target mosquito and the region. 

However,  the study lacks a little bit of innovation/originality in scientific research here.  Specific details in the study about behavior of the mosquito ( resting on a place to get attached to spores, seemed that 30 minutes are enough- on a sunny place (heating the wall) or in shadow..., or an investigation about the conidia/blastospores in the liquid for the larvae- are they going down on the ground (do blastospores behave here the same than a conidia suspension? where pupae able to develop than to adults or the prolonged larval duration had finally a lethal effect? ), just some ideas.. 

I have added some comments in the attached file. 

Best wishes! 

Reviewer 2 Report

Reviewer comments

Manuscript title: Entomopathogenic fungi as a potential management tool for the control of urban malaria vector, Anopheles stephensi (Diptera: 3 Culicidae)

Comments

The aim of this study was evaluation the pathogenicity of Beauveria bassiana and  M. anisopliae isolates against adult female An. stephensi on fungal treated cement and mud panels and to find out the impact of  fungal conidia and blastospores on larval development of An. stephensi.

-        The authors used isolated cultures of both tested fungi. Where are the references of these fungal isolates?, how do they identify? and where are their accession no?.

-        This manuscript lack the novelty. There are many references about the mosquitocidal activities of the tested fungi against An. stephensi

-        Line 302 (might be the first report on B. bassiana isolate that has virulence to An. stephensi adults)

It was reported before the mosquitocidal activity of  B. bassiana against adults of An. stephensi. Refer to the following articles.

·       Evaluation of the pathogenicity and infectivity of entomopathogenic hypocrealean fungi, isolated from wild mosquitoes in Japan and Burkina Faso, against female adult Anopheles stephensi mosquitoes. Fungal Ecology, Volume 15, June 2015, Pages 39-50.

·       Fitness consequences of larval exposure to  on- adults of the malaria vector Anopheles stephensi, Journal of Invertebrate Pathology, Volume 119, June 2014, Pages 19-24.

Therefore, my opinion is this manuscript is not suitable for publication Journal of Fungi.

Sincerely

Reviewer 3 Report

Line 109-110  To make it clear that t the isolates from soil or insect cadavars, which one is from the soil, which one is from insect cadavars.

Line122-123How about the filter paper, can those conidias can pass the filter paper.

To discuss the mechanism of the conidia and blastspores infeting the mosquitors in the discussion.

Round 2

Reviewer 1 Report

Dear Authors, 

Thanks for proofing comments and implementing those once into your manuscript. 

In general selection of the most appropriate strain (virulence, persistence, mass production, host specifity, etc..) when working with a new target insect should be standard. That study is in general not very innovative, but suitable for your local conditions and the specific target insect.  

Please just comment in one or two sentences why your study is innovative in that field (application of EPF on sites, where adults rest, developmental stages of the insect in water (specific issue for the pathogen) etc... 

Best wishes,

Reviewer 2 Report

The manuscript is Ok after the clarifications of the authors.
